# Analysis of the crystal structure of an active MCM hexamer

**Justin M Miller[†], Buenafe T Arachea[†], Leslie B Epling, Eric J Enemark***

Department of Structural Biology, St Jude Children's Research Hospital, Memphis, United States

**Abstract** In a previous Research article (*Froelich et al., 2014*), we suggested an MCM helicase activation mechanism, but were limited in discussing the ATPase domain because it was absent from the crystal structure. Here we present the crystal structure of a nearly full-length MCM hexamer that is helicase-active and thus has all features essential for unwinding DNA. The structure is a chimera of *Sulfolobus solfataricus* N-terminal domain and *Pyrococcus furiosus* ATPase domain. We discuss three major findings: 1) a novel conformation for the A-subdomain that could play a role in MCM regulation; 2) interaction of a universally conserved glutamine in the N-terminal Allosteric Communication Loop with the AAA+ domain helix-2-insert (h2i); and 3) a recessed binding pocket for the MCM ssDNA-binding motif influenced by the h2i. We suggest that during helicase activation, the h2i clamps down on the leading strand to facilitate strand retention and regulate ATP hydrolysis.

*For correspondence: eric.
enemark@stjude.org

[†]These authors contributed
equally to this work

Competing interests: The
authors declare that no
competing interests exist.

Reviewing editor: Michael R
Botchan, University of California,
Berkeley, United States

## Introduction

Hexameric MCM rings act as the replicative DNA helicase (*Bochman and Schwacha, 2008*; *Ilves et al., 2010*), encircling the leading strand DNA template at the replication fork (*Fu et al., 2011*). Mcm2-7 complexes are loaded (reviewed in *Remus and Diffley, 2009*) to encircle double-stranded DNA (dsDNA) via a 'gate' between Mcm2 and Mcm5 (*Bochman and Schwacha, 2007*, *2008*; *Costa et al., 2011*) to yield a double hexamer (*Evrin et al., 2009*; *Remus et al., 2009*) that does not unwind DNA. During helicase activation, the Dbf4-dependent Cdc7 kinase (DDK) and cyclin-dependent kinases (CDKs) drive recruitment of Cdc45 and the GINS complex (*Labib, 2010*). These factors stimulate the Mcm2-7 ATPase and helicase (*Ilves et al., 2010*) and with Mcm2-7 form the CMG complex (Cdc45-Mcm2-7-GINS), the active replicative helicase (*Moyer et al., 2006*; *Bochman and Schwacha, 2008*; *Ilves et al., 2010*). Following activation, two Mcm2-7 helicases encircle single-stranded DNA (ssDNA) and translocate independently (*Yardimci et al., 2010*), 3′→5′, on the leading strand DNA template (*Fu et al., 2011*) with the ATPase domain leading (*McGeoch et al., 2005*).

Our previous crystal structure of the *Pyrococcus furiosus* MCM N-terminal domain (*Pf*MCM$_N$) bound to ssDNA revealed an MCM single-stranded binding motif (MSSB) that binds ssDNA (*Froelich et al., 2014*). Our discussion of an MSSB role in helicase activation invoked action of the AAA+ (reviewed in *Duderstadt and Berger, 2013*) ATPase domain to translocate DNA, but we could not discuss specifically how the MSSB was affected by the AAA+ domain because it was not present in the *Pf*MCM$_N$:ssDNA structure. Now, we present the crystal structure of a helicase-active MCM hexamer to reveal a novel conformation for the A-subdomain that could play a role in MCM regulation and how the AAA+ helix-2-insert is tethered to the N-terminal domain to create a recessed binding pocket for the MSSB.

## Results and discussion

We identified a chimera of the N-terminal domain of *Sulfolobus solfataricus* (*Sso*) and the AAA+ domain of *Pf*MCM, *Sso-Pf*MCM (*Figure 1A*) with a robust DNA unwinding activity (*Figure 1B*; *Figure 1—figure supplement 1*). We present an analysis of the crystal structure of the *Sso-Pf*MCM hexamer bound to Mg/ADP (*Table 1*).

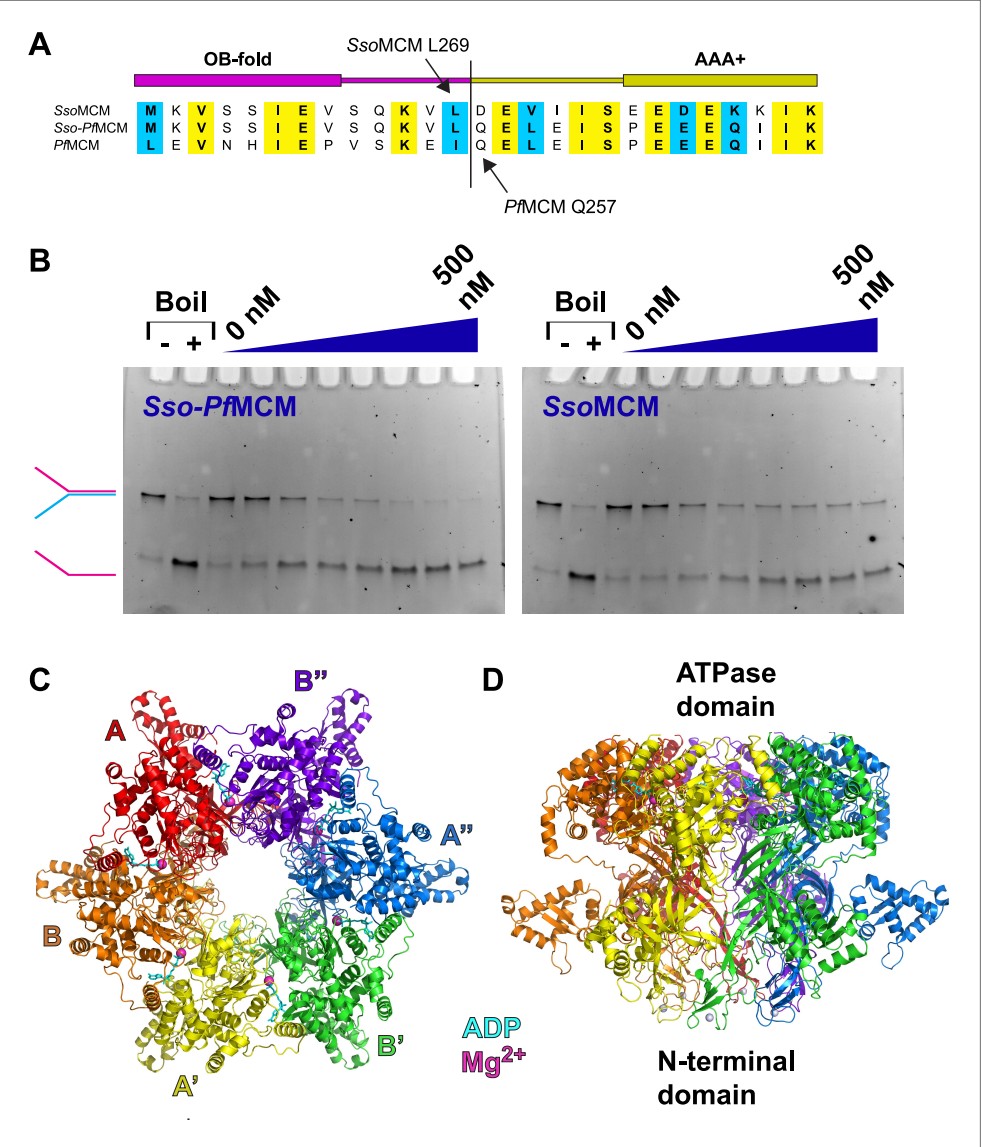

**Figure 1**. Properties of *Sso-Pf*MCM. (**A**) Sequence alignment showing the construction of the *Sso-Pf*MCM chimera. The N-terminal domain of *Sso*MCM (residues 1–269, top sequence) was fused to the AAA+ domain of *Pf*MCM (starting at residue 257, bottom sequence) to yield the chimera (middle sequence). (**B**) The *Sso-Pf*MCM chimera shows enhanced unwinding activity when compared to wild-type *Sso*MCM. Helicase reactions were performed at 69°C for 60 min with a Y-shaped DNA substrate with a 5'-fluorescein label on one strand. Unwinding reactions were in the presence of 4 mM ATP and contained 0, 50, 100, 150, 200, 300, 400, or 500 nM protein. Views of the *Sso-Pf*MCM hexamer crystal structure parallel (**C**) and perpendicular (**D**) to the central channel with each subunit uniquely colored. The magnesium ions are magenta spheres, and ADP molecules are shown as cyan stick. (**C**) View down the crystallographic threefold axis with the unique and symmetry-derived chains labeled. The ATPase domains are projected out of the page. (**D**) View perpendicular to the channel axis. The ATPase domains are located at the top, and the N-terminal domains are located at the bottom. The Zn ions are light grey spheres at the bottom.

The following figure supplements are available for figure 1:

**Figure supplement 1**. MCM catalyzed DNA unwinding visualized by gel electrophoresis.

**Figure supplement 2**. The relative positions of the N- and C-terminal domains in *Sso-Pf*MCM significantly differ from previous monomeric crystal structures.

**Figure supplement 3**. Activity and structure of *Pf*MCM$_{AAA}$.

**Table 1.** Data collection and refinement statistics

| | Sso-PfMCM:MgADP Hexamer | PfMCM_AAA:MgADP Double-octamer |
|---|---|---|
| Data collection | | |
| Space group | P6_3 | P1 |
| Cell dimensions | | |
| a, b, c (Å) | 118.902, 118.902, 199.317 | 124.956, 127.082, 128.025 |
| α, β, γ (°) | 90, 90, 120 | 71.852, 72.819, 80.392 |
| Resolution (Å) | 50–2.70 (2.80–2.70) | 50–3.80 (3.94–3.80) |
| $R_{sym}$ | 0.107 (0.750) | 0.169 (0.429) |
| $I/\sigma I$ | 14.8 (1.79) | 8.3 (2.41) |
| Completeness (%) | 99.8 (98.3) | 98.9 (97.0) |
| Redundancy | 6.8 (5.0) | 3.3 (2.6) |
| Refinement | | |
| Resolution (Å) | 50–2.70 (2.80–2.70) | 50–3.80 (3.94–3.80) |
| No. reflections | 39,044/1976 (2042/123) | 69,126/3486 (6206/357) |
| $R_{work}/R_{free}$ | 0.263/0.295 (0.360/0.353) | 0.301/0.314 (0.367/0.368) |
| No. atoms | | |
| Protein | 9432 | 2429 (1/16 of ASU) |
| ADP | 54 | 27 (1/16 of ASU) |
| ions | 10 | 1 (1/16 of ASU) |
| Water | 0 | 0 |
| B-factors | | |
| Protein | 60 | 91 |
| ADP | 118 | 72 |
| ions | 75 | 81 |
| Water | N/A | N/A |
| R.m.s. deviations | | |
| Bond lengths (Å) | 0.010 | 0.010 |
| Bond angles (°) | 1.488 | 1.597 |

## MCM:Mg/ADP hexamer crystal structure

The *Sso-Pf*MCM hexamer (*Figure 1C–D*; *Video 1*, 0:00) forms a ring with a channel large enough to accommodate double-stranded DNA (dsDNA) (see 'Materials and methods'). The *Sso-Pf*MCM structure therefore could mechanistically represent the structure of the MCM hexamer prior to loading (no DNA), after it loads to encircle dsDNA, or after its activation (encircling ssDNA). The two-tiered structure is consistent with electron microscopy studies (*Chong et al., 2000*; *Pape et al., 2003*; *Gomez-Llorente et al., 2005*; *Costa et al., 2006*; *Bochman and Schwacha, 2007*; *Remus et al., 2009*; *Costa et al., 2011*). The N-terminal tier has three subdomains, A–C (*Fletcher et al., 2003*; *Liu et al., 2008*; *Froelich et al., 2014*) with the A-subdomains in a different orientation (*Figure 2A*; *Video 1*, 0:20; see below) than observed previously. The relative positions of MCM_N and MCM_AAA differ considerably compared to monomeric and filament MCM crystal structures (*Brewster et al., 2008*; *Bae et al., 2009*; *Slaymaker et al., 2013*) (*Figure 1—figure supplement 2*). This difference is needed to prevent serious clashes that would occur among the ATPase domains.

## A-subdomain is in a different orientation than in previous structures

The A-subdomains of *Sso-Pf*MCM are rotated 150° compared to those in *Sso*MCM_N (*Liu et al., 2008*) (*Figure 2A*). The previously identified A-subdomain conformation seems fully possible in our present structure (see *Video 1*, 0:20), and we suggest that different A-subdomain conformations might play a role in MCM regulation. A similar A-subdomain rotation has been suggested for *Methanothermobacter*

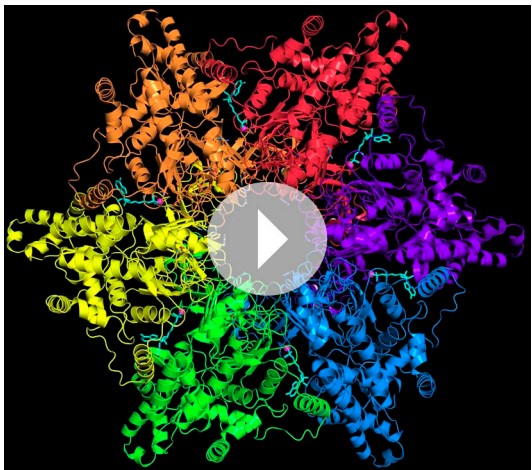

**Video 1**. Crystal structure details for *Sso-Pf*MCM. The video illustrates the arrangement of the subunits in the hexamer and the positions of the subdomains. The A-subdomain conformation is animated to transform to that observed in other crystal structures of MCM$_N$ to illustrate how they differ. The different A-subdomain conformations correlate with the conformation of a proline (P104) at the junction between the A- and C-subdomains. The relative position of the *mcm5-bob1* mutation is noted. Several central channel modules are highlighted, including the ps1β, h2i, β-turn, MSSB, and the interaction of ACL Q198 with the h2i. The ATPase site is compared to that of papillomavirus E1 (**Enemark and Joshua-Tor, 2006**), and several key residues are highlighted for MCM. The MSSB location is shown in a surface representation to illustrate that it sits at a recessed binding pocket where the ssDNA (green) of the aligned *Pf*MCM$_N$:ssDNA crystal structure (**Froelich et al., 2014**) would position snugly.

*thermautotrophicus* (*Mt*) MCM$_N$ based on electron microscopy (**Chen et al., 2005**) and was suggested to dictate helicase activity (**Chen et al., 2005**). We suggest the different conformations might regulate interaction with other factors: the A-subdomain might be stabilized against the OB-fold during one cell cycle stage, masking an interaction surface that becomes exposed by a conformational switch. In *Sso-Pf*MCM, the change correlates with a conformer change in P104 at the junction of the A/C-subdomains (phi/psi = −48.5/133.0 in *Sso-Pf*MCM vs −78.0/−14.9 in *Sso*MCM$_N$, **Liu et al., 2008**). This proline and an associated aromatic residue, F49 (**Figure 2A**; **Video 1**, 0:35), are conserved in Mcm2 and Mcm6 (**Video 1**, 0:47), suggesting these subunits could be particularly specialized conformational switches. Interestingly, the A-subdomain of Mcm2 interacts with Cdc45 (**Costa et al., 2011**), providing a potential link between A-subdomain conformation and CMG assembly. While we expect A-subdomain conformations to be more flexible without a proline, proline is not required to attain the conformation in our structure, and the corresponding residue in *Mt*MCM is a serine. As noted previously (**Chen et al., 2005**), A-subdomain rotation is a conceptual extension of the 'domain-push' mechanism described for *mcm5-bob1* (**Hardy et al., 1997**) where the A- and C-subdomain interaction is weakened by bulky side-chains (**Fletcher et al., 2003**). In our structure, as suggested for *Mt*MCM (**Chen et al., 2005**), the A/C subdomain interaction is not only weakened, it is broken altogether. Changes in A-subdomain conformation might be driven by MCM phosphorylation, such as phosphorylation of the *Sc*Mcm4 N-terminal serine/threonine-rich domain by DDK (**Sheu and Stillman, 2006**) that serves both inhibitory and facilitating roles in replication (**Sheu and Stillman, 2010**).

## Central channel modules

Several modules are directed into the central channel where they could interact with encircled DNA (**Figure 2B**; **Video 1**, 1:19). The pre-sensor-1-β-hairpin (ps1β) projects a universally conserved lysine, K785, that is essential for unwinding by *Sso*MCM (**McGeoch et al., 2005**). The helix-2-insert (h2i), required for helicase activity in *Mt*MCM (**Jenkinson and Chong, 2006**), prominently directs R734 and W741 into the central channel. These residues are conserved in a family-specific fashion for Mcms (**Video 1**, 1:32). The h2i and ps1β are located further from the N-terminal domain than predicted by monomeric crystal structures (**Figure 1—figure supplement 2**) due to interdomain differences (see above). The h2i projects further into the channel than the ps1β and appears to divide the AAA+ and N-terminal DNA-binding regions. As viewed in **Figure 2B**, the ps1β and the h2i direct their putative DNA-binding residues above the h2i, while the MSSB is below the h2i. The h2i creates a DNA-binding pocket at the MSSB where ssDNA was observed previously (**Froelich et al., 2014**). In *Sso*MCM, alanine mutants of lysine residues in this pocket (K129A and K194A) show severe DNA-binding and unwinding defects (**Pucci et al., 2004**). While the MSSB pocket of *Sso-Pf*MCM appears poised to bind ssDNA in the fashion observed previously (**Figure 2C**), some remodeling of h2i side-chains or the ssDNA would be necessary to avoid clashes, particularly involving F737.

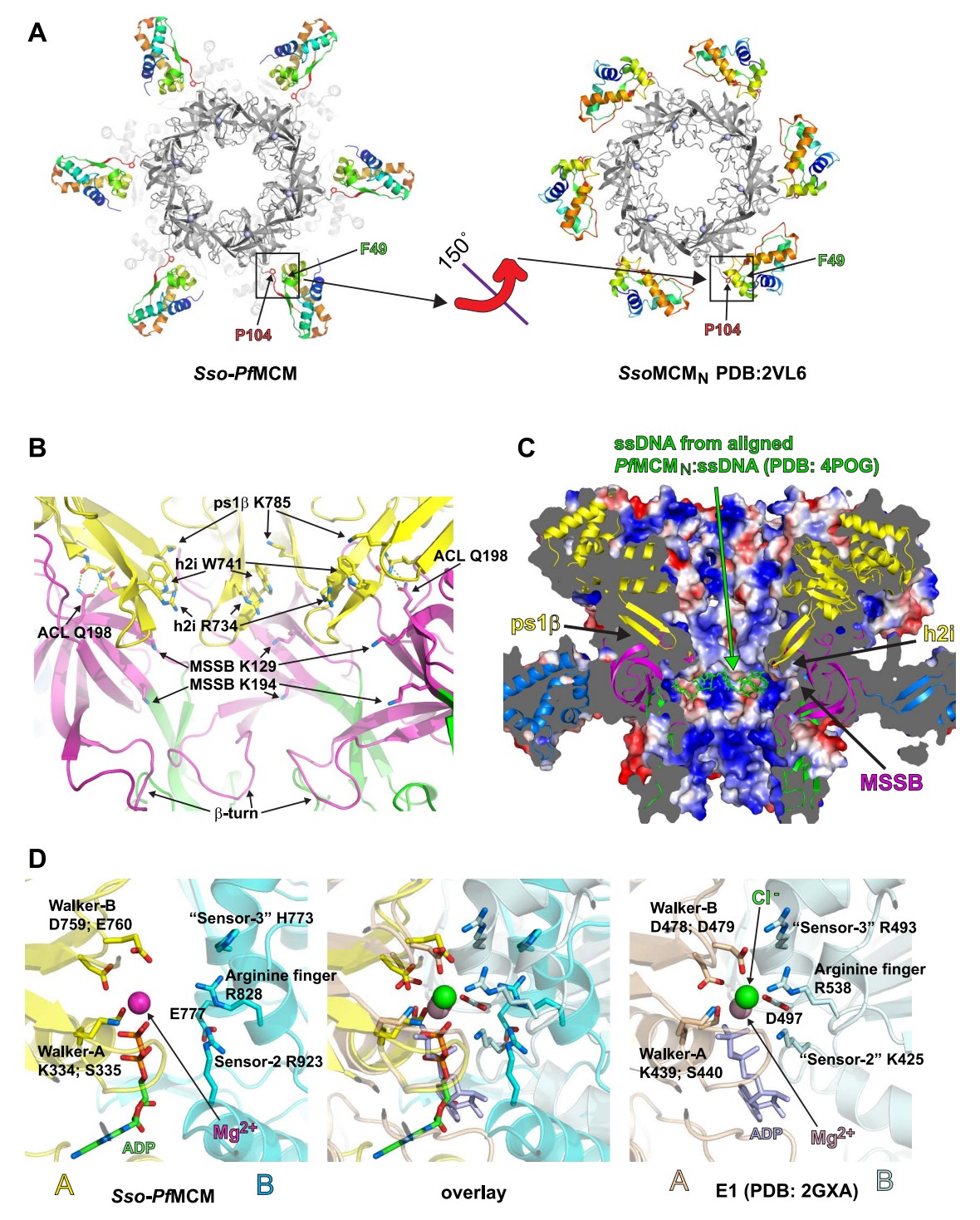

**Figure 2**. *Sso-Pf*MCM crystal structure details. (**A**) The A-subdomains (rainbow) of *Sso-Pf*MCM adopt a unique conformation that is rotated 150° compared to other crystal structures of hexameric MCM_N such as *Sso*MCM_N (***Liu et al., 2008***). The distinct conformations correlate with the conformation of P104 that is located at the junction between the A- and C-subdomains (boxed). In both conformations, P104 packs against the aromatic residue F49. (**B**) The modules of the central channel. The cartoon is colored with the AAA+ domain in yellow, the Zn-binding B-subdomain in green, and the OB-fold C-subdomain in magenta. The ps1β projects a conserved lysine, K785, into the channel. This lysine packs against W741 of the h2i, which sits adjacent to R734 of the h2i. The MSSB is recessed and sits below the h2i and above the β-turn. The N-terminal domain is tethered to the h2i by a universally conserved glutamine in the ACL, Q198. (**C**) Surface representation of *Sso-Pf*MCM colored by electrostatic potential. The surface is clipped
*Figure 2. Continued on next page*

*Figure 2. Continued*

with a vertical plane through the center to illustrate the central channel features. A cartoon representation of the protein with select modules labeled is colored as in *Figure 2B* with the helical A-subdomain in blue. The MSSB sits at a recessed pocket. The ssDNA from the aligned *Pf*MCM$_N$:ssDNA structure (*Froelich et al., 2014*) would be snugly positioned in this pocket. (**D**) Comparison of the *Sso-Pf*MCM ATPase site (left) with that of E1 (*Enemark and Joshua-Tor, 2006*, right). The Walker-A and Walker-B residues of one subunit (yellow) are positioned at the left side of the site while three positively charged residues of the adjacent subunit (cyan) line the right side of the site. An acidic residue of the cyan subunit sits below the site. Based on the superposition (middle), we predict that the MCM subunits need to approach each other more closely to generate a competent ATPase site.

The following figure supplement is available for figure 2:

**Figure supplement 1**. Cartoon model showing a role for an interaction between the N-terminal ACL conserved glutamine and the AAA+ h2i during helicase activation.

## Allosteric Communication Loop

In each subunit of our *Sso-Pf*MCM hexamer crystal structure, residues 198–212 comprise a conserved loop of the OB-fold that projects towards the h2i of the same subunit and the ps1β of an adjacent subunit. This loop has been termed the 'Allosteric Communication Loop' (ACL) (*Barry et al., 2009*) due to its predicted proximity to the AAA+ domain and its observed influence on unwinding (*Sakakibara et al., 2008*; *Barry et al., 2009*). Although the ATPase domain alone is sufficient to generate DNA unwinding in *Sso*MCM (*Barry et al., 2007*; *Pucci et al., 2007*) and in *Aeropyrum pernix* MCM (*Atanassova and Grainge, 2008*), several mutants located on this loop show unwinding defects (*Sakakibara et al., 2008*; *Barry et al., 2009*). In our hexamer structure, the proximity of the ACL to the h2i and ps1β, highly significant AAA+ modules (see above), strongly supports an ACL role in N- and C-terminal domain communication. The ACL position near the ps1β of an adjacent subunit is also consistent with previous studies (*Barry et al., 2009*).

We tested the DNA unwinding activity of *Pf*MCM$_{AAA}$ to compare with that of the *Sso-Pf*MCM chimera to explore interdomain communication. We found *Pf*MCM$_{AAA}$ had a negligible unwinding activity (*Figure 1—figure supplement 3*). The inactivity of *Pf*MCM$_{AAA}$ could result from attributes identified in the *Pf*MCM$_{AAA}$ double-octamer crystal structure (see 'Materials and methods'). Specifically, the non-hexameric ring architecture or the alternative topology for the helix-2-insert region, both of which we consider artifacts of the N-terminal truncation, could fully explain the lack of unwinding by this domain. Thus, although *Sso*MCM$_N$ does enhance the unwinding activity of *Pf*MCM$_{AAA}$, this could simply be due to enforcing a hexameric architecture or by disallowing the unusual h2i topology in favor of the canonical topology.

We next examined the *Sso-Pf*MCM structure for interdomain interactions that could be conserved in native proteins. The structure reveals a fully conserved glutamine, Q198, of the ACL interacts with the main-chain amide atoms of the h2i (*Figure 2B*; *Video 1*, 1:50). This interaction could occur in any MCM protein because the constituent atoms are fully conserved. The Q198:h2i interaction is not required for DNA unwinding because no unwinding defects are observed for the Q198A mutant (*Figure 1—figure supplement 1*), and in the corresponding alanine mutant of *Mt*MCM (*Sakakibara et al., 2008*). We suggest that the glutamine-h2i interaction functions prior to unwinding to lock the h2i in a holding position. The movement of DNA proposed previously to facilitate initial strand separation (*Froelich et al., 2014*) could occur by ATP-hydrolysis-driven inward movement of the h2i bound to DNA (*Figure 2—figure supplement 1*). After the h2i modules reach the position observed in our structure, they would be locked in place by interaction with Q198. This would tighten the grasp on one strand while the opposing strand exits the open Mcm2/5 gate (*Bochman and Schwacha, 2007*; *Costa et al., 2011*). With the h2i modules fixed in this position, the ATPase sites would be unable to adopt a productive hydrolysis conformation because each ATPase site is fundamentally tied to h2i position, potentially to prevent further ATPase activity during an important activation event. ATP hydrolysis inhibition by fixed h2i is directly analogous to inhibition of ATP hydrolysis in ϕ12 P4, an RNA-translocating hexamer, by cross-linked RNA-binding loops (*Kainov et al., 2008*).

## ATPase active site

The ATPase active site structurally resembles the ATPase site of the AAA+ helicase E1 (*Enemark and Joshua-Tor, 2006*) (*Figure 2D*, *Video 1*, 2:04) with Walker-A/B residues (*Abrahams et al., 1994*;

*Neuwald et al., 1999*) of one subunit, and three positively charged residues of the adjacent subunit. The three positive residues consist of sensor-2 (*Neuwald et al., 1999*), the arginine finger (*Neuwald et al., 1999*), and residues that we classify as sensor-3 (*Enemark and Joshua-Tor, 2006*). Although not typical for AAA+ proteins, the placement of sensor-2 in an ATPase site among Walker-A/B residues of the neighboring subunit (*in trans*) was predicted for MCM proteins in defining AAA+ Clade 7 (*Erzberger and Berger, 2006*). Biochemical experiments (*Moreau et al., 2007*) and MCM structure-based predictions (*Bae et al., 2009*) are also consistent with this arrangement. Based on comparison of the ATPase site with the tight 'ATP-like' configuration of E1 (*Enemark and Joshua-Tor, 2006*), we expect MCM subunits must approach more closely to generate a competent ATPase site. In this state, the h2i and ps1β are predicted to move upward in the view shown in *Figure 2—figure supplement 1* to increase the distance between the ACL and the ps1β as identified by DEER-spectroscopy (*Barry et al., 2009*). ATP-hydrolysis would drive the h2i/ps1β downward to translocate one ssDNA strand with expected polarity and orientation (*McGeoch et al., 2005*) while the complementary strand is excluded from the ring (*Fu et al., 2011*). Our present structural findings cannot differentiate several mechanistic details such as hydrolysis order or timing (reviewed in *Singleton et al., 2007*). We speculate that the MCM helicase unwinds DNA with helically-arranged h2i/ps1β modules analogous to E1 (*Enemark and Joshua-Tor, 2006*) and Rho (*Thomsen and Berger, 2009*), but the six non-identical subunits of Eukaryotic Mcm2-7 could operate asymmetrically during unwinding. Indeed, the ATPase modules of the different Mcm2-7 subunits show distinct roles and specialization during different functional stages (*Coster et al., 2014*; *Kang et al., 2014*). Elucidation of how the AAA+ domain interacts with DNA in an unwinding conformation will help reveal more details of the MCM unwinding mechanism.

## Materials and methods

### Cloning, mutagenesis, expression, and purification

The chimera protein construct consists of $SsoMCM_N$, ($SsoMCM$ aa 1–269) fused to $PfMCM_{AAA}$ ($PfMCM$ aa 257–361/729–966 = aa 257–966 with its intein, aa 362–728, removed). It corresponds to a full-length MCM protein lacking the short (aa 967–1049) C-terminal helix-turn-helix domain (*Aravind and Koonin, 1999*) that is dispensable for unwinding activity in *Mt*MCM (*Jenkinson and Chong, 2006*) and in *Sso*MCM (*Barry et al., 2007*). All expression constructs were prepared as N-terminal $His_6$-SUMO fusions. The original SUMO vector was the generous gift of Dr Christopher D Lima (*Mossessova and Lima, 2000*). The *Pf*MCM gene contains an intein, aa 362–728 (*Yoshimochi et al., 2008*) in the helix-2-insert region of ATPase domain. We genetically removed the intein by sequentially cloning two fragments (amplified from *P. furiosus* genomic DNA, ATCC) incorporating a silent NotI mutation at the junction. This plasmid served as the PCR template to generate constructs of *Sso-Pf*MCM (*Sso*MCM aa 1–269/*Pf*MCM aa 257–361/729–966, pJM001.3) and $PfMCM_{AAA}$ (aa 263–361/729–966, pEE021.1 = crystallized construct; and *Pf*MCM aa 252–361/729–1049 = construct of unwinding experiments). The *Sso-Pf*MCM chimera construct was generated by overlap extension of PCR fragments encoding the N-terminal domain of *Sso*MCM (amplified from *S. solfataricus* genomic DNA, ATCC) and $PfMCM_{AAA}$ that had 93 bases of overlap at the junction. The full-length *Sso*MCM expression construct (pEE045.1) was prepared via PCR amplification of the full *Sso*MCM gene. The Q198A mutant (pJM005.5) was generated by site-directed mutagenesis of pJM001.3. DNA sequencing verified the integrity of the coding region of all constructs. Proteins were purified as described previously (*Froelich et al., 2014*), including removal of the SUMO tag by digestion by Ulp1 protease (the Ulp1 protease plasmid was the generous gift of Dr Christopher D Lima) (*Mossessova and Lima, 2000*).

### Crystallization, data-collection, structure-solution, and refinement

Prior to crystallization, purified *Sso-Pf*MCM was dialyzed into buffer containing 25 mM HEPES, pH 7.6; 10 mM NaCl; and 5 mM Mg(OAc)$_2$. Crystals of Sso-*Pf*MCM with Mg/ADP grew by hanging drop by mixing 2 µl of protein:ADP solution (10.8 mg/ml *Sso-Pf*MCM; 5 mM ADP) and 2 µl of well solution (100 mM HEPES, pH 7.6; 350 mM MgCl$_2$; 3% (wt/vol) PEG 3350). Crystals were cryoprotected by quickly passing through a 1:3 ethylene glycol:well solution and flash frozen in liquid nitrogen. Data were collected at SER-CAT beamline 22-ID. Data were collected at 1.0 Å wavelength in 0.25° oscillations for 112.5° at a temperature of 100 K. All data were scaled and integrated using the HKL-2000 software package (*Otwinowski and Minor, 1997*) to 2.70 Å resolution.

The structure was solved in space group P6$_3$ by the program Phaser (**McCoy et al., 2007**), which placed two copies of *Pf*MCM$_{AAA}$ (see below) and two copies of a monomer of *Sso*MCM$_N$, PDB 2VL6 (**Liu et al., 2008**), in a single hexamer on a crystallographic threefold axis. Overall, the unit cell contains two nearly sixfold symmetric hexamers offset by a strong NCS translation of [1/3, 2/3, 1/2] (Patterson peak height 50% of origin). Initial electron density maps revealed a clear misplacement of the helical A-subdomain, which was corrected by Phaser (**McCoy et al., 2007**) by using separate search models for the A-subdomain and the B/C-subdomains. The initial electron density map was greatly improved by multi-crystal electron density averaging in the AAA+ domain region with the program Dmmulti (**Cowtan, 1994**) by implementing twofold averaging of *Sso-Pf*MCM and 16-fold averaging of *Pf*MCM$_{AAA}$. The model was refined at various stages with CNS (**Brunger et al., 1998**; **Brunger, 2007**), phenix (**Afonine et al., 2012**), and refmac5 (**Vagin et al., 2004**) and manually improved with Coot (**Emsley and Cowtan, 2004**). The final refinement was carried out in CNS (**Brunger et al., 1998**; **Brunger, 2007**). A Ramachandran plot calculated by Procheck (**Laskowski et al., 1993**) indicated the following statistics: core: 917 (86.8%); allowed: 124 (11.7%); generously allowed: 15 (1.4%); disallowed: 0 (0%). Figures were prepared with PyMOL (**Schrodinger, 2010**), Molscript (**Kraulis, 1991**), and Raster3D (**Merritt and Bacon, 1997**).

Crystallographic datasets for crystals grown in the presence of several nucleotide cofactors (ADP, AMP-PNP, ATP-γS, ADP-AlF$_x$) were collected. All were strongly isomorphic with crystals grown with Mg/ADP, and no evidence of a γ-PO$_4$ or its analog was ever detected in resulting electron density maps. We therefore conclude that all crystal datasets adopt a highly similar structure most consistent with an ADP-bound state and that the γ-PO$_4$/analog hydrolyzes over the course of crystallization or is crystallographically disordered.

The crystal structure of *Pf*MCM$_{AAA}$ was pivotal in obtaining the crystal structure of *Sso-Pf*MCM (above), and we therefore include the details of its structure determination. However, the resolution of *Pf*MCM$_{AAA}$, (3.80 Å) limits the overall detail of the structure. Crystals of *Pf*MCM$_{AAA}$ with Mg/ADP grew by hanging drop by mixing 2 μl protein/Mg/ADP (6 mg/ml; 5 mM ADP; 50 mM MgCl$_2$; 18 mM HEPES, pH 7.6; 180 mM NaCl; 4.5 mM β-mercaptoethanol) with 2 μl well solution (50 mM sodium cacodylate, pH 6.0; 50 mM magnesium acetate; 30% MPD; 5% glycerol). A Crystal was flash frozen in liquid nitrogen and data were collected at SER-CAT beamline 22-BM at 1.0 Å wavelength in 0.5° oscillations for 360° at a temperature of 100 K. The data were scaled and integrated using the HKL-2000 software package (**Otwinowski and Minor, 1997**) to 3.80 Å resolution. A weak molecular replacement solution was obtained with the program Phaser (**McCoy et al., 2007**), which placed 16 monomers of the AAA+ portion of PDB 4FDG (**Slaymaker et al., 2013**) as a double-octamer. The MCM complex is not presumed to adopt an octameric assembly in vivo, but we note that a hypothetical hexamer generated by removing two adjacent subunits from the octameric ring would correspond to an open hexameric ring with an opening large enough to permit entry of B-form dsDNA. Initial electron density maps were greatly improved by 16-fold NCS-averaging and solvent flattening with the program Resolve (**Terwilliger, 2000**, **2004**), which revealed obvious side-chain positions. The sequence was assigned to the structure based upon the location of selenium positions (7 per subunit) for a selenomethionine derivative in an NCS-averaged anomalous difference Fourier map generated with the Resolve-improved phases, and by alignment with the 1.90 Å resolution structure of a monomeric MCM homolog (**Bae et al., 2009**). The selenomethionine derivative was expressed in B834(DE3) cells (EMD Millipore, Darmstadt, Germany) in LeMaster's media (**Hendrickson et al., 1990**), and anomalous difference data were collected at SER-CAT beamline 22-ID at 0.97915 Å wavelength in 0.5° oscillations for 260° at a temperature of 100 K. The data were scaled and integrated using the HKL-2000 software package (**Otwinowski and Minor, 1997**) to 4.0 Å resolution. The anomalous signal was too weak to generate starting phases, but readily identified the selenium positions by an NCS-averaged anomalous difference fourier map generated with the Resolve-improved molecular replacement phases (see above). The structure was refined with a strict 16-fold NCS protocol in CNS (**Brunger et al., 1998**; **Brunger, 2007**) and manually improved with Coot (**Emsley and Cowtan, 2004**). Following refinement of the higher resolution structure of *Sso-Pf*MCM (above), the coordinates were updated, and the h2i region was rebuilt. The structure was subjected to coordinate and group B-factor refinement with strict 16-fold NCS in CNS (**Brunger et al., 1998**; **Brunger, 2007**) to yield the final model. The h2i is folded differently than in other AAA+ proteins to mediate a β-sheet interface with the ps1β of a subunit in the other octamer. Twofold symmetric dimers are arranged around an eightfold symmetry axis to yield approximate D8-symmetry. A total of 16 of these β-sheet interfaces occur around the ring. This h2i structure is not compatible with the position of the N-terminal domain seen in *Sso-Pf*MCM, and it is

therefore almost certainly an artifact of removing the N-terminal domain in the $Pf$MCM$_{AAA}$ construct. The precise sequence registry for these h2i residues is not clearly defined, and the residues have been modeled as poly-alanine in our best assessment of the polypeptide direction. A Ramachandran plot calculated by Procheck (*Laskowski et al., 1993*) indicated the following statistics: core: 213 (74.7%); allowed: 56 (19.6%); generously allowed: 8 (2.8%); disallowed: 8 (2.8%).

## Helicase assay

All unwinding experiments were performed with a Y-shaped DNA substrate with a 55-mer double-stranded region, a 50-mer poly-dT 3'-arm, and a 30-mer 5'-arm with a fluorescein label at the 5'- end. The substrate was prepared by annealing a 5'-fluorescein-labeled oligonucleotide (5'-TTGAACCA CCCCCTTGTTAAATCACTTCTACTTGCATGCCTGCAGGTCGACTCTAGAGGATCCCCGGGT ACCGAGCTCGAATTCG–3' with an unlabeled oligonucleotide (5'- CGAATTCGAGCTCGGTACCC GGGGATCCTCTAGAGTCGACCTGCAGGCATGCAAGTTTTTTTTTTTTTTTTTTTTTTTTTTTTTTTTT TTTTTTTTTTTTTTTT-3'), Sigma–Aldrich, St. Louis, MO). The 85-mer oligonucleotide was identical to a previously published substrate that had been annealed to M13 plasmid ssDNA to study $Sso$MCM unwinding (*Pucci et al., 2004*). Helicase activity assays were prepared in 20 µl reaction mixtures with 25 mM HEPES (pH = 7.6), 100 mM Na(OAc), 5 mM Mg(OAc)$_2$, 4 mM ATP, and 3.7 nM fluorescein-labeled DNA substrate. For protein concentration titrations, the protein concentration ranged from 0 to 500 nM (monomer), and reactions were incubated at 69°C for 60 min. For time-course experiments, 500 nM protein was incubated at 69°C for 1 to 90 min. Prior to the addition of ATP to initiate unwinding, time points were incubated at 69°C for 5 min to allow for thermal equilibration. For all samples, reactions were stopped by the addition of 5 µl of loading buffer containing 40% (vol/vol) glycerol, 5% (wt/vol) sodium dodecyl sulfate (SDS), and 50 mM ethylenediaminetetraacetic acid (EDTA), and a 20 µl aliquot was loaded on a 4–20% 1× TBE gradient PAGE gel (Biorad, Berkeley, CA) and run at 150 V for 90 min. Gel imaging was performed with a Fuji LAS-4000 using a 15-min exposure time and a SYBR-Green filter.

## Definition of central channel axis, mathematical analysis

We conclude that the central channel of our structure of the $Sso$-$Pf$MCM hexamer is sufficiently large to accommodate dsDNA based on a channel radius minimum of 12.7 Å for its polyalanine model. For comparison, the polyalanine model of BPV E1:ssDNA (encircles ssDNA; *Enemark and Joshua-Tor, 2006*; *Lee et al., 2014*) has a minimum radius of 6.1 Å (PDB: 2GXA, hexamer 1, *Enemark and Joshua-Tor, 2006*); topoisomerase I:dsDNA (tightly encircles dsDNA) has a minimum radius of 7.5 Å (PDB: 1A35, *Redinbo et al., 1998*); and PCNA (encircles dsDNA but can slide) has a minimum radius of 16.0 Å (PDB: 1PLQ, *Krishna et al., 1994*). Based on these comparisons, the channel diameter in our structure of $Sso$-$Pf$MCM is large enough to accommodate dsDNA, but might not slide over dsDNA as readily as PCNA. The details of these calculations are provided below.

Our analysis of the central channel of a ring requires a definition of the channel axis. We define this axis as the rotation axis for least-squares permutation of the subunits. For $Sso$-$Pf$MCM, this axis coincides with a crystallographic threefold axis. For straightforward analysis, we use the following procedure, which is general, to produce a transformed PDB coordinate file with the channel axis coincident with the [0, 0, Z] axis of the standard PDB coordinate system. First, the coordinates of a full ring molecule were translated to place the center-of-mass on the origin of the standard PDB coordinate system with the program MOLEMAN (*Kleywegt, 1997*). Second, the least-squares rotation that explicitly permutes the subunits (for example, chains ABCDEF onto chains BCDEFA in a hexamer) was calculated by LSQMAN (*Kleywegt, 1996*), expressed in polar angles (Omega = O, Phi = P, Chi = C). These polar angles were used to transform the origin-shifted coordinates in two successive polar rotation operations with the program MOLEMAN (*Kleywegt, 1997*). The first polar rotation was by (0, 0, −P), and the second was by (90, 90, −O). Subsequently, the distance of any atom to the channel axis can be calculated from its X and Y coordinates in the transformed PDB coordinate file as the square root of ($X^2$ + $Y^2$). For Topoisomerase I (PDB id: 1A35, *Redinbo et al., 1998*), the channel axis was defined as the least-squares helical axis of the bound dsDNA, and this axis was transformed to coincide with the [0, 0, Z] axis of the PDB coordinate system as described above.

## Acknowledgements

Data were collected at Southeast Regional Collaborative Access Team (SER-CAT) 22-ID and 22-BM beamlines at the Advanced Photon Source, Argonne National Laboratory. Supporting institutions may

be found at www.ser-cat.org/members.html. We are grateful to SER-CAT staff for experimental support. Use of the Advanced Photon Source was supported by the U. S. Department of Energy, Office of Science, Office of Basic Energy Sciences, under Contract No. W-31-109-Eng-38.

## Additional information

### Funding

| Funder | Grant reference number | Author |
|---|---|---|
| American Lebanese Syrian Associated Charities (ALSAC) | | Eric J Enemark |
| National Institute of General Medical Sciences | R01GM098771 | Eric J Enemark |
| Cancer Center Support Grant | 5 P30 CA021765 | Eric J Enemark |

The funders had no role in study design, data collection and interpretation, or the decision to submit the work for publication.

### Author contributions

JMM, Acquisition of data, Analysis and interpretation of data, Drafting or revising the article; BTA, Acquisition of data, Analysis and interpretation of data, Drafting or revising the article, Contributed unpublished essential data or reagents; LBE, Drafting or revising the article, Contributed unpublished essential data or reagents; EJE, Conception and design, Analysis and interpretation of data, Drafting or revising the article

## Additional files

### Supplementary file

• Supplementary file 1. ClustalW sequence alignment of several MCM proteins in MSF format. Portions of this alignment are shown throughout *Video 1*. The *Sso-Pf*MCM residue numbers in the alignment run consecutively, which is consistent with database numbering for the *Sso*MCM portion. For the *Pf*MCM portion, database numbers can be obtained from the aligned *Pf*MCM sequence. The intein (residues 362–728) was manually removed from the *Pf*MCM sequence for the alignment. Thus, for residues of *Pf*MCM in the alignment with sequence numbers greater than 361, 367 must be added to agree with database (intein-containing) sequence numbering. Residue numbering throughout the manuscript and *Video 1* agrees with the database numbering for the respective portions.

### Major dataset

The following dataset/s was/were generated:

| Author(s) | Year | Dataset title | Dataset ID and/or URL | Database, license, and accessibility information |
|---|---|---|---|---|
| Miller JM, Arachea BT, Epling LB, Enemark EJ | 2014 | Crystal structure of an active MCM hexamer (SsoPfMCM hexamer) | http://www.pdb.org/pdb/explore/explore.do?structureId=4R7Y | Publicly available at RCSB Protein Data Bank. |
| Miller JM, Arachea BT, Epling LB, Enemark EJ | 2014 | PfMCM-AAA double-octamer | http://www.pdb.org/pdb/explore/explore.do?structureId=4R7Z | Publicly available at RCSB Protein Data Bank. |

The following previously published dataset/s was/were used:

| Author(s) | Year | Dataset title | Dataset ID and/or URL | Database, license, and accessibility information |
|---|---|---|---|---|
| Enemark EJ, Joshua-Tor L | 2006 | Crystal structure of papillomavirus E1 hexameric helicase with ssDNA and MgADP | http://www.pdb.org/pdb/explore/explore.do?structureId=2gxa | Publicly available at RCSB Protein Data Bank. |

| Redinbo MR, Stewart L, Kuhn P, Champoux JJ, Hol WG | 1998 | Human topoisomeras I/DNA complex | http://www.pdb.org/ pdb/explore/explore. do?structureId=1a35 | Publicly available at RCSB Protein Data Bank. |
|---|---|---|---|---|
| Krishna TS, Kong XP, Gary S, Burgers PM, Kuriyan J | 1994 | Crystal structure of the eukaryotic DNA polymerase processivity factor PCNA | http://www.pdb.org/ pdb/explore/explore. do?structureId=1plq | Publicly available at RCSB Protein Data Bank. |
| Liu W, Pucci B, Rossi M, Pisani FM, Ladenstein RLiu W, Pucci B, Rossi M, Pisani FM, Ladenstein R | 2008 | Structural analysis of the Sulfolobus solfataricus MCM protein n- terminal domain | http://www.pdb.org/ pdb/explore/explore. do?structureId=2vl6 | Publicly available at RCSB Protein Data Bank. |
| Slaymaker IM, Fu Y, Toso DB, Ranatunga N, Brewster A, Forsburg SL, Zhou ZH, Chen XS | 2013 | Crystal Structure of an Archaeal MCM Filament | http://www.pdb.org/ pdb/explore/explore. do?structureId=4fdg | Publicly available at RCSB Protein Data Bank. |

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
