## [Decision Letter]

Thank you for sending your work entitled “Analysis of the crystal structure of an active MCM hexamer” for consideration at *eLife.* Your article has been favorably evaluated by James Manley (Senior editor) and 3 reviewers, one of whom is a member of our Board of Reviewing Editors.

The following individuals responsible for the peer review of your submission have agreed to reveal their identity: Michael Botchan (Reviewing editor), and James Berger and Alessandro Costa (peer reviewers).

The Reviewing editor and the other reviewers discussed their comments before we reached this decision, and the Reviewing editor has assembled the following comments to help you prepare a revised submission.

This Research advance provides the first crystal structure of an Mcm hexamer and the use of a chimera fusing the N-terminal domain of one archeal Mcm to the AAA+ domain of another to achieve the goal is novel. The structure provides some surprises in particular that the A domain in the N-terminal tier undergoes a big shift in conformation from the monomer to the hexamer. For now this discovery is a heuristic one and the ideas that such a switch might be regulatory in going from one inactive state to another and active is intriguing. That the P104 residue which undergoes a big Psi/Phi change from one state to another is conserved (though there is no need for it) in the eukaryote Mcm 2 and 6 feeds the speculation especially since this could affect how Cdc45 interacts with Mcm2 in the CMG. We feel that this advance is what we are aiming for in *eLife* to add something important to an already published piece. And this advance does show how the amino tier might be affected by the full deal. However, the following points need to be addressed in a revised manuscript before publication.

1) The helicase assay is somehow a bit disappointing and we would like to see the intact unwound substrate actually enter into the gel and not be hung up in the well. This would give one assurances that the circle is intact and that the displaced oligo is somehow not an artifact of a nuclease that might require ATP. But even more important would be a side by side comparison of this full hexamer with the AAA+ domain itself which should also be active. Would the full length hexamers be more active than the AAA+ tier alone? That might help the manuscript and support the interaction ideas put forward. Surely the authors have that hexamer in hand to compare. Mutations in a few key residues might also test easily a few points if these can be tested in the time required for a re-submission. Further the reviewers ask for this specifically.

A time-course assay and an enzyme concentration titration need to be performed, and these data should be compared side-by-side with wild-type SsoMCM or PfuMcm. If the helicase activity of the chimera is severely compromised in relation to the native enzymes, then the significance attributed to certain structural states/interactions (such as the h2i-MSSB conformation) may have to be revised.

2) We agree that the idea that the different A-domain conformations might regulate interactions with other factors or control helicase function is intriguing, however, this concept is not tested. In general the impact of the glutamine-h2i interaction on MCM activities is not examined, nor is it clear how the current h2i conformation might (or might not) control ssDNA binding by the MSSB. At least some degree of follow-up that better relates the structural findings to MCM function is needed before publication in *eLife* can be recommended.

Specifically these points must be addressed: concerning the statement that “…during helicase activation, the h2i clamps down on the leading strand to facilitate strand retention and prevent premature ATP hydrolysis”: it is hard to see how this idea might work. The MMSB binds ssDNA in a manner that is perpendicular to the axis of any dsDNA duplex that might thread its way through the central helicase pore following loading. However, ssDNA isn't formed until after (or perhaps commensurate with) any activation event. Moreover, ATP hydrolysis by the MCMs has been reported to be important for activation (and hence ssDNA formation). This concept needs to be more clearly articulated or revised.

3) Concerning the statement, “The relative positions of MCMN and MCMAAA differ considerably compared to crystal structures of monomeric MCM”: how so? These differences could be shown/discussed more clearly to better highlight what is new here compared to Brewster et al. or Bae et al.

---

## [Author Response]

*1) The helicase assay is somehow a bit disappointing and we would like to see the intact unwound substrate actually enter into the gel and not be hung up in the well. This would give one assurances that the circle is intact and that the displaced oligo is somehow not an artifact of a nuclease that might require ATP. But even more important would be a side by side comparison of this full hexamer with the AAA+ domain itself which should also be active. Would the full length hexamers be more active than the AAA+ tier alone? That might help the manuscript and support the interaction ideas put forward. Surely the authors have that hexamer in hand to compare. Mutations in a few key residues might also test easily a few points if these can be tested in the time required for a re-submission. Further the reviewers ask for this specifically*.

*A time-course assay and an enzyme concentration titration need to be performed, and these data should be compared side-by-side with wild-type SsoMCM or PfuMcm. If the helicase activity of the chimera is severely compromised in relation to the native enzymes, then the significance attributed to certain structural states/interactions (such as the h2i-MSSB conformation) may have to be revised*.

The Sso-PfMCM chimera shows an unwinding activity that appears faster than that of wild-type SsoMCM (see below). We therefore feel that this is a very good model system for studying the structure of the MCM hexamer.

Manufacturer literature indicates that M13 plasmid DNA is too large to enter the lowest percentage TBE gel available, and we therefore adopted a smaller Y-shaped substrate for the unwinding experiments. The intact form of this Y-shaped substrate readily enters the gel and does not hang up in the well. We have performed protein concentration titrations and time-course assays for Sso-PfMCM, full-length SsoMCM, and Sso-PfMCM Q198A. Protein concentration assays that compare Sso-PfMCM with SsoMCM are now presented in Figure 1, and we present the results for the 3 proteins for side-by-side comparison as Figure 1—figure supplement 1. We have also performed a comparable protein concentration titration for the AAA+ domain of PfMCM, which showed negligible activity. We have chosen to present the lack of activity for this domain alongside the double-octamer crystal structure of the AAA+ domain of PfMCM in order to underscore potential reasons for the inactivity (Figure 1—figure supplement 3). Specifically, the non-hexameric ring architecture or the alternative topology for the helix-2-insert region, both of which we consider artifacts of the N-terminal truncation, could fully explain the lack of unwinding by this domain. We think it is important to emphasize that these possibilities underlie the lack of unwinding activity rather than imply direct contradiction with previous reports that the AAA+ domains of SsoMCM (7, 51) and ApeMCM (4) are sufficient for unwinding activity. While it would be accurate to say that the N-terminal domain of SsoMCM enhances the unwinding activity of PfMCM-AAA+, this could simply be due to enforcing a hexameric architecture or by disallowing the unusual h2i topology (at the interfaces of the octamers of the PfMCM-AAA+ double-octamer) in favor of the canonical topology (as we observe in Sso-PfMCM).

*2) We agree that the idea that the different A-domain conformations might regulate interactions with other factors or control helicase function is intriguing, however, this concept is not tested. In general the impact of the glutamine-h2i interaction on MCM activities is not examined, nor is it clear how the current h2i conformation might (or might not) control ssDNA binding by the MSSB. At least some degree of follow-up that better relates the structural findings to MCM function is needed before publication in* eLife *can be recommended*.

*Specifically these points must be addressed: concerning the statement that “…during helicase activation, the h2i clamps down on the leading strand to facilitate strand retention and prevent premature ATP hydrolysis”: it is hard to see how this idea might work. The MMSB binds ssDNA in a manner that is perpendicular to the axis of any dsDNA duplex that might thread its way through the central helicase pore following loading. However, ssDNA isn't formed until after (or perhaps commensurate with) any activation event. Moreover, ATP hydrolysis by the MCMs has been reported to be important for activation (and hence ssDNA formation). This concept needs to be more clearly articulated or revised*.

The A-domain conformation was a complete surprise because all previous crystal structures had shown a consistent, but different, conformation. We were further intrigued by a plausible mechanistic similarity to the well-known mcm5-bob1 mutation. We agree that at this stage, the possibility of alternative A-domain configurations playing a role in MCM regulation/activity is not tested, but we feel that the definitive demonstration of an alternative A-domain orientation in a MCM hexamer is sufficiently intriguing to be described in detail.

We now show that mutation of the conserved glutamine (Q198A) shows no defect in the unwinding activity of Sso-PfMCM (Figure 1—figure supplement 1), consistent with a previous study showing no unwinding defects in the alanine mutant of the corresponding glutamine of MtMCM (56). Along with select glycines conserved for the OB-fold itself, this glutamine is one of the most conserved residues in the MCM N-terminal domain, and yet we observe no role in the unwinding activity. We therefore feel this residue likely has a universal role in an activity other than unwinding, such as helicase activation.

Our description of a possible role of the h2i/Q198 interaction during helicase activation was confusing. This was intended to show how the AAA+ domain might participate in the activation mechanism suggested in Figure 7 of the earlier manuscript. We now show this in Figure 2—figure supplement 1 with a similar color scheme as in the earlier manuscript. We certainly believe that ATP hydrolysis would occur initially, but that ATP hydrolysis would cease after the h2i has reached the position identified in our structure and becomes tethered to Q198. We suggest that this would be a holding position for the complex until an important event, such as extrusion of the complementary strand, has completed.

We feel that the figure helps clarify the relative motions that we describe for the two domains. We intend to remain neutral on several aspects of helicase mechanism. This led us to initially describe the roles of the protein modules vaguely. The design of Figure 2—figure supplement 1 could be interpreted to suggest that all 6 h2i/ps1b modules collectively move in a coplanar fashion, which could carry an implication of a concerted ATP hydrolysis mechanism. We don’t intend to imply these features, but also do not intend to rule them out. The present structure does not provide a basis to discriminate among several mechanistic scenarios (reviewed in [60]), which we state in the text.

*3) Concerning the statement, “The relative positions of MCMN and MCMAAA differ considerably compared to crystal structures of monomeric MCM”: how so? These differences could be shown/discussed more clearly to better highlight what is new here compared to Brewster et al. or Bae et al*.

We now show these differences explicitly in Figure 1—figure supplement 2 and describe these differences in greater detail the figure legend. For both cases, the overall hexameric complex that is generated by superimposing six monomers onto each N-terminal domain OB-fold would generate serious intermolecular clashes among the AAA+ domains.